# Thermal Behavior and Pyrolysis Kinetics of Mushroom Residue with the Introduction of Waste Plastics

**DOI:** 10.3390/polym15183824

**Published:** 2023-09-19

**Authors:** Jiale Li, Tao Pu, Zhanghong Wang, Taoze Liu

**Affiliations:** 1College of Eco-Environmental Engineering, Guizhou Minzu University, Guiyang 550025, China; 15885794651@163.com (J.L.);; 2Engineering Research Center of Green and Low-Carbon Technology for Plastic Application, Guizhou Minzu University, Guiyang 550025, China

**Keywords:** edible fungi residue, waste plastics, co-pyrolysis, thermal behavior, pyrolysis kinetics

## Abstract

Co-pyrolysis is considered a very promising technology for the treatment of solid wastes as it can rapidly realize the volume reduction of raw materials and obtain high value-added products. To realize the resource utilization of newly emerging solid wastes in relation to edible fungi residue and waste plastics, mushroom residue (MR), a representative of edible fungi residue, was co-pyrolyzed with waste plastic bags (PE), waste plastic lunch boxes (PP), and waste plastic bottles (PET). The thermal behavior and pyrolysis kinetics of the mixtures were investigated. It was found that the softening of the plastics in the mixtures led to an increase in the initial pyrolysis temperature of MR by 2–27 °C, while the pyrolytic intermediates of MR could greatly promote the decomposition of the plastics, resulting in a decrease in the initial pyrolysis temperatures of PE, PP, and PET in the mixtures by 25, 8, and 16 °C, respectively. The mixture of MR and PE (MR/PE) under different mixture ratios showed good synergies, causing the pyrolysis peaks attributed to MR and PE to both move towards the lower temperature region relative to those of individual samples. The increase in heating rate led to enhanced thermal hysteresis of the reaction between MR and PE. The strength of the interaction between plastics and MR based on mass variation was subject to the order PE > PP > PET. The pyrolysis activation energies of MR, PE, PP, and PET calculated from kinetic analysis were 6.18, 119.05, 84.30, and 74.38 kJ/mol, respectively. The activation energies assigned to MR and plastics were both reduced as plastics were introduced to co-pyrolyze with MR, indicating that MR and plastics have a good interaction in the co-pyrolysis process. This study provides theoretical and experimental guidance for the resource utilization of agricultural solid wastes via thermochemical conversion.

## 1. Introduction

The edible fungi industry is an important pillar industry to ensure China’s food security and economic development. According to statistics, the annual production of edible fungi in China can reach 96.05 million tonnes [1]. Concomitantly, edible fungi residue, the largest byproduct of the edible fungi industry, is produced at five times the amount of edible fungi every year [2,3]. Meanwhile, edible fungi residue is rich in protein, fat, minerals, and other nutrients, which can easily breed pathogenic bacteria, and is an important source of soil pollution and water pollution [4]. Currently, the main methods to treat edible fungi residue mainly include incineration, composting, making animal feed, making flower substrate, and reuse to cultivate edible fungi [2,5]. Overall, there are some shortcomings to these methods such as long treatment cycles, low utilization efficiency, and secondary environmental problems [6]. There is an urgent need to develop reasonable ways to recycle edible fungi residue, which will be key to promoting the green and sustainable development of the edible fungi industry.

On the other hand, plastics are produced and used in large quantities every day as an important material for people’s lives. It is reported that global plastic production reached 367 million tonnes in 2020 and is expected to reach 710 million tonnes by 2040 [7]. Plastics are relatively stable, difficult to biodegrade, and highly mobile in the environment, so that the recovery rate of plastic waste is only 10%. In total, 76% of waste plastics eventually enter landfills or natural ecosystems and the remaining 14% are disposed of by incineration [8]. Waste plastic landfill occupies a large area, has high treatment costs, and can produce toxic and harmful substances [9]. The incineration of waste plastics can produce a large amount of greenhouse gases and highly toxic substances such as dioxins, furans, and polychlorinated biphenyls, causing global warming and a variety of human diseases [10,11]. Therefore, it is of great significance to explore green resource utilization methods for plastics.

Pyrolysis technology is commonly used in the treatment of solid wastes, such as agricultural and forestry waste, sludge, livestock and poultry manure, and municipal solid waste, due to its ability to rapidly realize the volume reduction and thermal transformation of raw materials and obtain high value-added products [12]. However, in the pyrolysis process of solid waste, especially biomass-based solid waste, the high oxygen content and complex composition of the raw materials lead to relatively poor quality of the products obtained [13]. For example, previous studies have shown that bio-oils derived from biomass pyrolysis generally have the disadvantages of low calorific value, high oxygen content, high corrosiveness, and instability [14]. In contrast, co-pyrolysis technology involves mixing raw materials with different physical and chemical properties and then carrying out pyrolysis. Co-pyrolysis technology can realize the simultaneous treatment of multi-component raw materials. Meanwhile, due to the existence of good synergy between different components of the raw materials in the process of pyrolysis, co-pyrolysis technology greatly improves the quality of the resulting products [15]. Currently, co-pyrolysis technology is widely used in the synergistic treatment of biomass and sludge, biomass and coal, biomass and plastics, etc. [15,16,17]. Among these, plastics are considered to be the ideal raw material for co-pyrolysis with biomass due to their ability to release large numbers of hydrogen radicals and hydrocarbon radicals during the pyrolysis process, which promotes the deoxygenation of biomass and the breaking of chemical bonds, strengthens the decomposition of biomass, and obtains large amounts of products of good quality [18]. For example, Chen et al. found that the melt coating effect of plastic can greatly promote the pyrolysis of microalgae and significantly reduce the activation energy of microalgae during co-pyrolysis [19]. Similarly, when bamboo was co-pyrolyzed with plastic at a mass ratio of 1:3, an excellent synergistic effect was observed according to thermogravimetric analysis [20].

Therefore, to achieve rapid and efficient resource utilization of edible fungi residue, it is proposed to treat edible fungi residue by adding waste plastics for co-pyrolysis. Currently, related studies are relatively limited and the mechanism of the co-pyrolysis process is still unclear. Accordingly, this study took mushroom residue (MR) as a typical representative of edible fungi residue and blended it with plastic for pyrolysis. Conditions relating to the types of plastics (PE, PP, PET), the ratio of MR to plastics (5:1–1:5), and heating rate (10–40 °C/min) during the co-pyrolysis process were controlled. Thermogravimetric analysis (TGA) and kinetic models were used to analyze the pyrolysis parameters, interactions, and kinetic parameters of the pyrolysis process. This research will provide important theoretical guidance for the resource utilization of agricultural solid waste.

## 2. Materials and Methods

### 2.1. Raw Materials and Sample Preparation

Discarded edible fungi sticks after the cultivation of edible fungi were collected from a mushroom cultivation base in Majiang County, Guizhou Province, China. The sticks were dried at 60 °C for 12 h and then pulverized and sieved through a 40-mesh sieve (below 0.38 mm). After that, mushroom residue powder (MR) was obtained. Waste plastics were collected from the waste recycling station in Huaxi University Town, Guiyang, China, including three types of waste plastics, plastic bags (polyethylene, PE), plastic lunch boxes (polypropylene, PP), and plastic bottles (polyethylene terephthalate, PET) (see Appendix A). The collected waste plastics were cleaned and baked in an oven at 60 °C for 12 h. Subsequently, the plastics were sheared to 0.2–2 mm powder. Then, 3 g of MR and 3 g of PE were weighed and put in an agate mortar. After grinding for 10 min, the mixed sample of MR and PE was obtained, denoted as MR/PE-1:1 where 1:1 indicates the mass ratio of MR and PE in the sample. Based on the same method, mixed samples of specific mass ratios were prepared separately (see Appendix A).

### 2.2. Thermogravimetric Analysis

The pyrolytic behavior of the samples was analyzed using a thermogravimetric analyzer (TGA) (Netzsch-Feinmahltechnik GmbH, Selb, Germany). About 10 mg of the sample was placed in a corundum crucible and heated from room temperature to 800 °C at different heating rates (10–40 °C/min) under an N_2_ gas flow of 50 mL/min, and the change in weight loss of the sample during the process was recorded. The weight loss (TG) and the rate of weight loss (DTG) of the sample with increasing temperature were calculated according to Equations (1) and (2). More than two parallel runs of each experiment were performed to ensure the accuracy of the results.
(1)$$ŋ=100\%\;\times \;\frac{W_{0}-W_{\mathrm{temperature}}}{W_{0}}$$
(2)$$\varphi =\frac{dŋ}{\mathrm{dt}}$$
where W_0_ and W_temperature_ is the initial mass of sample and that at temperature T, respectively (mg); t is the time corresponding to the temperature (min).

### 2.3. Kinetic Study

The pyrolytic properties of samples can be quantitatively described by kinetic analysis using kinetic parameters such as activation energy (E) and pre-exponential factor (A). It is generally recognized that the pyrolysis of biomass, such as coal, plastics, and agricultural wastes is consistent with a first-order reaction [21]. This can be expressed as the following equation (Equation (3)):(3)$$\frac{\mathrm{dx}}{\mathrm{dt}}=A\;\exp\;(-\frac{E}{\mathrm{RT}})(1-\alpha )$$
where E is activation energy (kJ/mol); A is pre-exponential factor (1/min); T is temperature (K); R is a universal gas constant (J/mol K); α is conversion of raw material which can be calculated from the following equation (Equation (4)):(4)$$\alpha =\frac{W_{0}-{\;W}_{t}}{W_{0}-{\;W}_{f}}\times 100\%$$
where W_0_ is the initial mass of raw material (mg); W_t_ is the mass of raw material at a certain time (mg); W_f_ is the mass at the termination of the reaction (mg).

When the heating rate H (H = dT/dt) of pyrolysis is constant, Equation (3) can be integrated as the following equation (Equation (5)):(5)$$\ln\left[\frac{-\ln\left(1-\alpha \right)}{T^{2}}\right]=\ln\left[\frac{\mathrm{AR}}{\mathrm{HE}}\left(1-\frac{2\mathrm{RT}}{E}\right)\right]-\frac{E}{\mathrm{RT}}$$

Due to RT ≪ 1, ln[AR/HE (1 − 2RT/E)] can accordingly be considered as a constant. According to Equation (5), E and A can be calculated by the slope ln[–ln (1 – α)/T^2^] and intercept 1/T, respectively.

### 2.4. Evaluation of the Synergistic Interaction of MR and Plastics

The difference in weight loss (∆W) between the theoretical and actual values was used to express the degree of synergism between MR and plastics, which can be calculated from the following equation (Equation (6)):(6)$$∆{W=W}_{\mathrm{mixture}}-{(x}_{1}W_{1}{+x}_{2}W_{2})$$
where W_mixture_ is the weight loss of the mixtures, which is obtained from the TG data; x_1_W_1_ + x_2_W_2_ is the calculated theoretical weight loss of the corresponding mixture, where x_1_ and x_2_ are the corresponding mass fractions of MR and plastics (PE, PP and PET), respectively; W_1_ and W_2_ are the corresponding weight loss of MR and plastics, respectively. In principle, ∆W is close to zero if there is no interaction between the individual fractions.

## 3. Results and Discussion

### 3.1. Thermogravimetric Analysis of Individual Component

The TG and DTG curves for MR and plastics (PE, PP, PET) pyrolyzed separately are shown in Figure 1, and corresponding parameters are shown in Table 1. Pyrolysis of MR consists of three stages, i.e., dehydration stage, main pyrolysis stage, and carbonization stage. The pyrolysis temperature interval of the first stage was 30–201 °C with a mass loss of 13%, which is mainly attributed to the removal of free water and crystalline water from MR. With the increase in temperature, the MR began to deeply remove crystalline water and induced the components with poor thermal stability in the structure for glass transition [22]. The pyrolysis zone of the second stage was 201–497 °C, and the mass loss was 40%, which is mainly related to the rapid decomposition of organic components such as cellulose, hemicellulose, and part of the lignin in the MR. As a result, a large amount of volatile matter escapes and a sharp decrease in mass can be seen [18]. The pyrolysis interval of the last stage was 497~800 °C with a mass loss of 9%. The pyrolysis intermediates of MR are further decomposed or condensed at high temperatures to form carbon products accompanied by a slight mass decline [23]. The MR produced 26.96% of solid residue after pyrolysis. The DTG result shows that the main pyrolysis stage of MR presents a broad and blunt weight loss peak, with peak temperature and maximum weight loss rate of 342 °C and −0.46%/min, respectively. The shoulder peak of the main pyrolysis stage can be attributed to the decomposition of hemicellulose, while the tailing phenomenon is related to the fact that lignin has a wide decomposition interval [24].

Compared with MR, the pyrolysis processes of PE and PP are relatively simple, with only two pyrolysis stages. For PE, the temperature intervals of the first and second stages are room temperature–416 and 416–503 °C, respectively. Due to its low moisture content and good thermal stability, only structural softening occurs at lower temperatures and the mass loss is negligible. As the pyrolysis temperature increases (>416 °C), PE decomposes quickly with almost no solid residue. The mass loss is more than 99%. The DTG result shows that PE presented a narrow but strong weight loss peak at this stage, and the maximum pyrolysis weight loss rate and the corresponding temperature were −3.01%/min and 480 °C, respectively, which are highly related to the relatively low fixed carbon content and high volatile matter content of PE [25]. Similar to PE, the main pyrolysis temperature range of PP is 341~492 °C, with a weight loss of 96%, and the maximum pyrolysis weight loss rate and corresponding temperature are −2.56%/min and 460 °C, respectively. The relatively narrow weight loss peaks of PE and PP are mainly because they are both hydrocarbon polymers. The structural units and the linkage bonds of the polymer are relatively simple, and there is almost no solid residue after decomposition [12]. In contrast, the pyrolysis process of PET is relatively complex because its structural units are mainly composed of aromatic rings and oxygen-containing functional groups, and the polymer linkage bonds are complex [26]. The pyrolysis of PET consists of three stages, i.e., room temperature—390, 390–494, and 494–796 °C. The first stage is mainly related to water loss and structural softening, resulting in a slight mass loss. With the increase in pyrolysis temperature (390–494 °C), PET undergoes a fierce decomposition with a weight loss of 81%. This is mainly attributed to the depolymerization of PET into monomers, accompanied by a small number of irregular chain-breaking reactions that release a large number of volatile components [27]. The remaining material is further decomposed in the third stage with a weight loss of 3%, and 14.98% solid residue is collected at the end. The maximum weight loss rate for PET pyrolysis is −1.95%/min at 445 °C. The initial pyrolysis temperature of the main pyrolysis interval of PE is 31 and 26 °C higher than those of PP and PET, respectively, indicating that the stability of PE is higher compared with those of PP and PET (see Table 1). It is worth noting that the main pyrolysis temperature intervals of PE, PP, and PET completely overlap with that of MR, which provides a good condition for the interaction between MR and plastics during co-pyrolysis [28].

### 3.2. Influence of Plastic Types on the Pyrolytic Behavior of MR

The TG and DTG curves of MR pyrolysis with the introduction of PP, PE, and PET (MR/PE, MR/PP, and MR/PET) are shown in Figure 2. The results show that there is an important effect of plastic types on the pyrolysis of MR. The pyrolysis of MR–plastic mixtures generally exhibits two main stages. The pyrolysis intervals of the first main stage of MR/PE, MR/PP, and MR/PET were 203–391 °C, 228–377 °C, and 219–374 °C, respectively, and this stage is mainly attributed to the rapid decomposition of hemicellulose, cellulose, and part of lignin of MR in the mixtures (see Appendix A). The initial temperatures of the main pyrolysis stages attributed to MR in MR/PE, MR/PP, and MR/PET were 2–27 °C higher than that of MR alone, indicating that the introduction of plastics greatly inhibited the pyrolysis of MR. This may be related to the structural softening of the plastics at a lower temperature. The softening plastics wrap around the MR particles, which hinder the heat and mass transfer of the reaction process as well as the release of the volatile components of MR [29]. The second main pyrolysis stage intervals of MR/PE, MR/PP, and MR/PET were 391–505 °C, 377–496 °C, and 374–615 °C, respectively, which is attributed to the fierce decomposition of the remaining lignin of the MR and plastics in the mixtures. The initial pyrolysis temperatures of this stage were 25, 8 and 16 °C lower than those of PE, PP, and PET pyrolysis alone, respectively, indicating that co-pyrolysis of plastics with MR can promote the pyrolysis of plastics to move into the low temperature zone. This is mainly because the intermediates produced by MR pyrolysis, especially oxygenated compounds, are highly reactive and capable of attacking the chemical bonds in the plastics and promoting their decomposition [30]. The solid residues of MR/PE, MR/PP, and MR/PET after decomposition were 5.64%, 11.47% and 21.29%, respectively, of which MR/PE and MR/PP were lower than corresponding theoretical values (13.66 and 13.42%, respectively). This indicates that the introduction of PE and PP can promote the deep decomposition of MR. In contrast, the solid residue after MR/PET pyrolysis was slightly higher than the theoretical value (20.97%). This proves that there is an interaction between MR and PET in the pyrolysis process, which promotes the generation of solid carbon products. In addition, the DTG curves show that the maximum weight loss temperatures attributed to MR and plastics in MR/PE, MR/PP, and MR/PET were 339, 341, 341 °C and 481, 467, 433 °C, respectively. These results shows that the temperature of the maximum weight loss rate attributed to MR decomposition in mixtures is slightly low (1–3 °C) compared with MR alone, while plastics show a different trend. In particular, the maximum weight loss rate temperature attributed to PET in the MR/PET mixture is reduced by 13 °C compared to PET pyrolysis alone.

### 3.3. Influence of Mixing Ratio on Pyrolytic Behavior of MR/PE

The TG-DTG curves of MR/PE with different mass ratios (5:1–1:5) are shown in Figure 3. As mentioned above, the pyrolysis of MR/PE can be divided into two main stages, which are attributed to the decomposition of MR and PE, respectively. In the first main stage, the weight losses of MR/PE-5:1, MR/PE-2:1, MR/PE-1:1, MR/PE-1:2, and MR/PE-1:5 were 29.16%, 9.63%, 14.13%, 6.20%, and 1.64%, respectively, which generally shows a decreasing trend with the increasing percentage of PE in the mixture (see Table 1). Compared with MR pyrolysis alone, the initial temperature of MR pyrolysis in the mixtures increased by 2–65 °C, which is partly related to the proportion of PE in the mixtures. This further confirms that the presence of PE in the mixtures inhibits the decomposition of MR. With the increase in the proportion of PE in the mixtures, the thermal hysteresis phenomenon of MR caused by the softening of PE to wrap the MR becomes more evident [29]. In the second main pyrolysis stage, the initial pyrolysis temperatures attributed to PE pyrolysis in MR/PE-5:1, MR/PE-2:1, MR/PE-1:1, MR/PE-1:2, and MR/PE-1:5 decreased by 36, 31, 25, 7, and 17 °C, respectively, in comparison with that of PE alone. This generally shows a positive correlation with the proportion of MR in the mixture. As aforementioned, the oxygenated intermediates produced by MR pyrolysis during MR/PE pyrolysis can promote the linkage breaking of PE and reduce the energy demand. With the increase in biomass proportion in the mixture, more oxygenated intermediates are produced, which can comprehensively attack the chemical linkages in the PE, thus promoting the rapid decomposition of the PE [31]. The solid residues of MR/PE-5:1, MR/PE-2:1, MR/PE-1:1, MR/PE-1:2 and MR/PE-1:5 were 18.17%, 16.30%, 5.64%, 8.70%, and 3.48%, respectively, which are lower than theoretical ones (22.53%, 18.09%, 13.66%, 9.22%, and 4.78% respectively). This reveals that there is significant synergistic interaction between MR and PE, irrespective of the ratio of MR to PE in the mixture. The DTG results show two consecutive major weight loss peaks corresponding to the decomposition of MR and PE, respectively, and the corresponding intensities are closely related to the mass ratio of MR and PE in the mixture (see Table 1). With the decrease in the ratio of MR to PE in the mixtures from 5:1 to 1:5, the maximum weight loss rate of MR in the mixtures decreased from −0.28%/min to −0.03%/min, while those of PE increased from −1.16%/min to −2.89%/min. According to the temperature corresponding to the maximum weight loss rate, the peaks corresponding to both MR and PE in the mixtures were found to move to the low-temperature region in comparison with single-component pyrolysis. Especially for PE, the temperature corresponding to the maximum weight loss rate obeys the order MR/PE-5:1 < MR/PE-2:1 < MR/PE-1:1 < MR/PE-1:2 < MR/PE-1:5, which indicates that good synergistic effects are maintained between MR and PE at different mass ratios.

### 3.4. Effect of Heating Rate on Pyrolytic Behavior of MR/PE

Figure 4 shows the TG-DTG curves of MR/PE at different heating rates (10–40 °C/min). It can be seen that all MR/PE at different heating rates show two evident weight loss stages, corresponding to the decomposition of MR and PE, respectively. The initial pyrolysis temperatures attributed to MR and PE in the mixtures increased from 203 and 391 °C to 236 and 419 °C, respectively, when the heating rate increased from 10 to 40 °C/min. This phenomenon is mainly attributed to the fact that there is a certain degree of delay in the heat and mass transfer at higher heating rates [32]. When the heating rate was 10, 20, 30, and 40 °C/min, the solid residues of MR/PE at 800 °C were 5.64%, 6.01%, 5.23%, and 9.74%, respectively, which generally shows a increased tendency with the increase in heating rate. This demonstrates that a higher heating rate is unfavorable to the deep decomposition of MR and PE, and weakens the interaction between MR and PE. The DTG curves show that the intensity of the maximum weight loss rate attributed to the MR in the mixtures was enhanced with the increase in the heating rate, while those attributed to PE showed a opposite trend. The temperatures corresponding to the maximum weight loss rates show that an increase in the heating rate (>10 °C/min) shifted the pyrolysis of MR and PE in the mixtures to a higher temperature region. 

### 3.5. The Interaction of MR and Plastics during Co-Pyrolysis Process

The variation of ∆W with the increase in pyrolysis temperature for MR/PE, MR/PP and MR/PET is shown in Figure 5. It is reported that ∆W greater or less than zero implies that there is a synergy between the individual components [18]. The ΔW values of MR/PE increased gradually (>0) with the increase of pyrolysis temperature in the temperature range from room temperature to 424 °C, indicating that the mass loss of MR/PE was severely hindered in this temperature range. The ∆W of MR/PE reached the maximum value of 17.22% at 424 °C. According to the TG result in Figure 1, this is mainly related to the fact that PE starts to soften and then encapsulate MR particles at lower temperature, resulting in a great inhibition of the release of MR/PE moisture and MR pyrolysis volatiles [33,34]. When PE starts to pyrolyze, the wrapping effect of PE is weakened, and large amounts of volatiles are released from the MR/PE, leading to a rapid decrease of ΔW to a negative value. The lowest value of −10.52% is found at 656 °C. The negative values of ΔW indicate that the interaction between MR and PE at the current temperature leads to the release of large amounts of volatiles from the mixture. It is worth noting that the MR/PE maintained relatively low ΔW values towards the end of the reaction (800 °C), indicating that a good synergistic interaction existed between MR and PE and their pyrolysis intermediates. This is consistent with the result that the solid residue of MR/PE is significantly lower than the theoretical one [30,35]. Similar to MR/PE, the ΔW of MR/PP also showed a tendency to increase first and then decrease, reaching its maximum value of 11.99% and minimum value of −2.13% at 460 and 682 °C, respectively. The difference between MR/PE and MR/PP is that the ΔW values of MR/PP were less than those of MR/PE throughout the pyrolysis process.

In contrast, the ΔW values of MR/PET showed a trend of decreasing first and then increasing, reaching the minimum value of −8.23% and the maximum value of 4.19% at 416 and 490 °C, respectively. The ΔW of MR/PET in the initial stage decreased gradually with the increase of pyrolysis temperature, which may be caused by the fact that the softening and wrapping effect of PET for MR particles is not crucial in comparison with PE and PP. On the contrary, a small amount of PET particles may be taken away during the release of MR moisture and volatiles, which affects the mass variation of the MR/PET mixture. When the pyrolysis temperature increased to 416 °C, softening and viscosity of the PET gradually appeared, affecting the mass change of the mixture. However, because the PET has already begun to decompose at this stage, the release of volatiles from MR/PET is hence slightly inhibited. It is worth noting that the ΔW values of MR/PET tend to zero when the pyrolysis temperature is >600 °C. This demonstrates that the interaction between MR and PET does not cause a significant mass change in this zone. Accordingly, the interactions between MR and plastics summarized on the basis of ΔW variation are presented in Figure 6.

### 3.6. Kinetic Analysis

The first-order reaction kinetic model was used to fit In[−ln(1 – α)/T^2^] versus 1/T for the main decomposition stages of MR, PE, PP, PET, and their mixtures, and the results are shown in Figure 7 and Table 2. The correlation coefficients (R^2^) of all samples are larger than 0.94, indicating that the adopted reaction model can well describe the corresponding pyrolysis processes. Among them, the pyrolysis of MR and individual plastics can be described by a single one-stage reaction, while the pyrolysis of the mixtures is fitted by two successive first-order reactions. The E values of MR, PE, PP, and PET were 6.18, 119.05, 84.30, and 74.38 kJ/mol, respectively, similar to previous studies [36,37,38]. The E values of plastics are 12~19 times higher than that of MR, indicating that plastics have relatively high thermal stability. This shows good consistency with the TGA results. For plastics, the strength of thermal stability based on E values is in the order of PE > PP > PET, which coincides with the temperatures corresponding to their maximum weight loss rates.

When MR was blended with plastics for pyrolysis, the E values attributed to MR in MR/PE, MR/PP, and MR/PET were reduced by 31.23–46.44% compared with that of MR alone, and those attributed to PE, PP, and PET were reduced by 33.48%, 33.76%, and 73.16% in comparison with those of plastics alone, respectively. This result suggests that the interactions of MR and plastics are conducive to reducing each other’s E values. When MR and PE were mixed at a mass ratio of 5:1 (MR/PE-5:1), the E value attributed to MR in the mixture was slightly increased by 9.87% compared with that of MR alone, while the E value attributed to PE was significantly decreased by 47.99%. The increase in the E value of MR in the mixture may be related to the softening and wrapping effect of PE, and the decrease in the E value of PE is attributed to the generation of a large number of oxygenated intermediates under a larger proportion of MR. It has been reported that the portion of biomass in mixture during co-pyrolysis has a positive effect on promoting the linkage breaking of PE [39]. With the decrease in the mass ratio of MR and PE, the E values attributed to MR decreased while the E values assigned to PE increased. It is worth noting that the E values of MR and PE during co-pyrolysis are generally lower than the pyrolysis of the corresponding single component (regardless of the mass ratio), which further confirms that the interaction of MR and PE is beneficial in lowering the E values of each other. There were irregular fluctuations in the E values attributed to MR and PE in the MR/PE mixtures at different heating rates. Similar results were obtained by Wang et al. when the effect of heating rate on the E values of LG and PE in LG/PE mixtures were investigated [39].

## 4. Conclusions

The effect of the addition of waste plastics (PE, PP, PET) on the thermal behavior and pyrolysis kinetics of MR was studied by TGA. Results show that the pyrolysis of MR consists of three stages, i.e., the dehydration stage, the main pyrolysis stage, and the dehydration stage. Among them, the temperature interval of the main pyrolysis stage is 201–497 °C, and the mass loss is 40%. When plastics were added to co-pyrolyze with MR, the initial pyrolysis temperatures in the mixtures assigned to MR increased by 2~27 °C due to the softening effect of the plastics. On the other hand, the intermediates of MR pyrolysis greatly promoted the decomposition of plastics, leading to a decrease in the initial pyrolysis temperatures of PE, PP, and PET in the mixtures by 25, 8, and 16 °C, respectively. When MR and PE were mixed in different mass ratios (5:1–1:5), all MR and PE showed good synergistic interaction. This caused the pyrolysis peaks attributed to MR and PE to move to the low-temperature region relative to the individual samples. When the heating rate was increased from 10 to 40 °C/min, the thermal hysteresis of the reaction between MR and PE was gradually enhanced. The ∆W results show that the strength for the interaction between MR and plastics was subject to the order PE > PP > PET. When plastics were introduced to co-pyrolyze with MR, the E values of MR and plastics were reduced, indicating that MR and plastics interacted well during the co-pyrolysis processes.

## Figures and Tables

**Figure 1 polymers-15-03824-f001:**
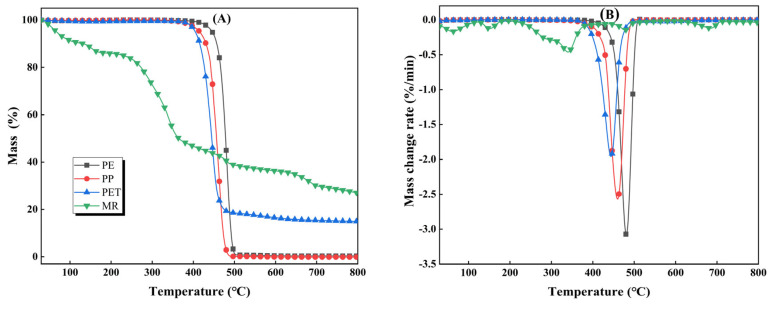
Pyrolytic behavior of MR, PE, PP, and PET: (**A**) TG curves; (**B**) DTG curves.

**Figure 2 polymers-15-03824-f002:**
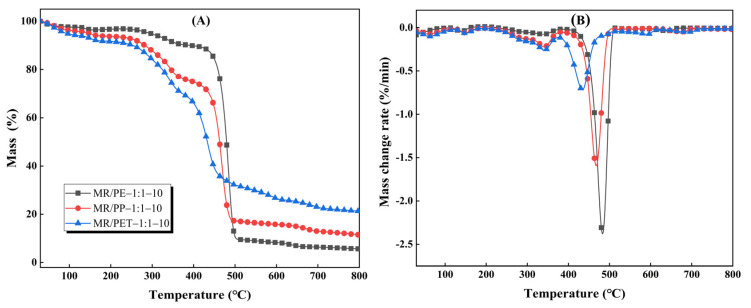
Pyrolysis characteristics of MR with the addition of PE, PP, and PET: (**A**) TG curves; (**B**) DTG curves.

**Figure 3 polymers-15-03824-f003:**
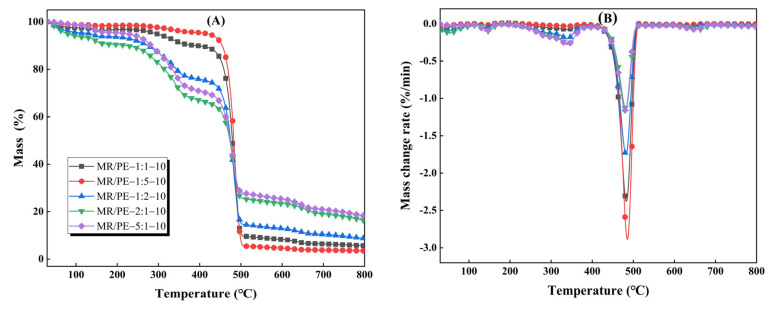
Characteristics of MR/PE co-pyrolysis under different mixing ratios: (**A**) TG curves; (**B**) DTG curves.

**Figure 4 polymers-15-03824-f004:**
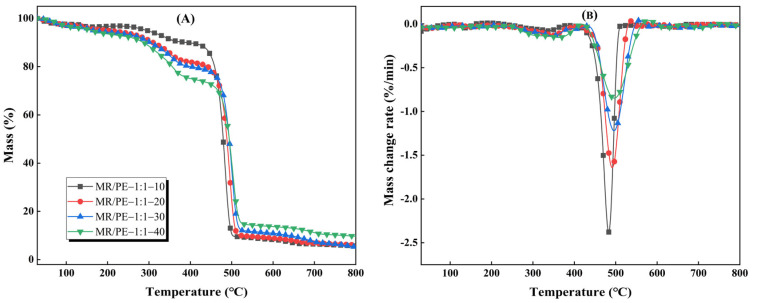
Characteristics of MR/PE co-pyrolysis under different heating rates: (**A**) TG curves; (**B**) DTG curves.

**Figure 5 polymers-15-03824-f005:**
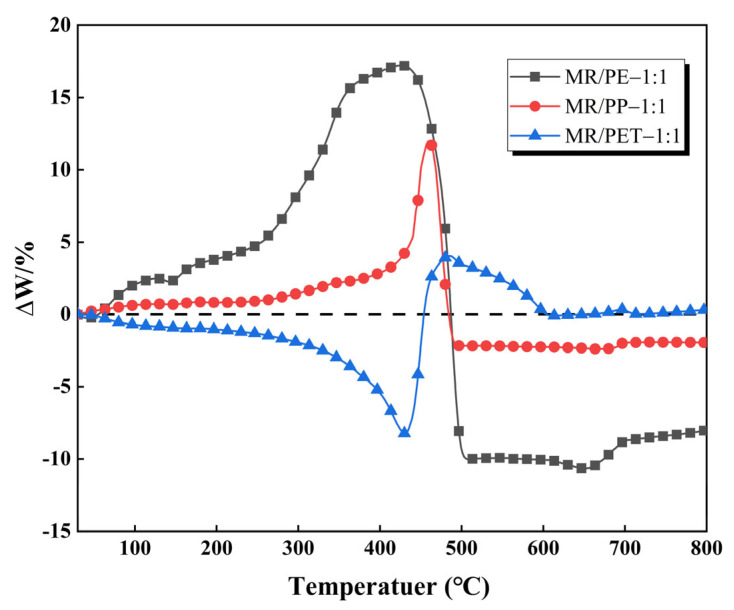
Variation of ∆W with pyrolysis temperature for MR/PE, MR/PP and MR/PET.

**Figure 6 polymers-15-03824-f006:**
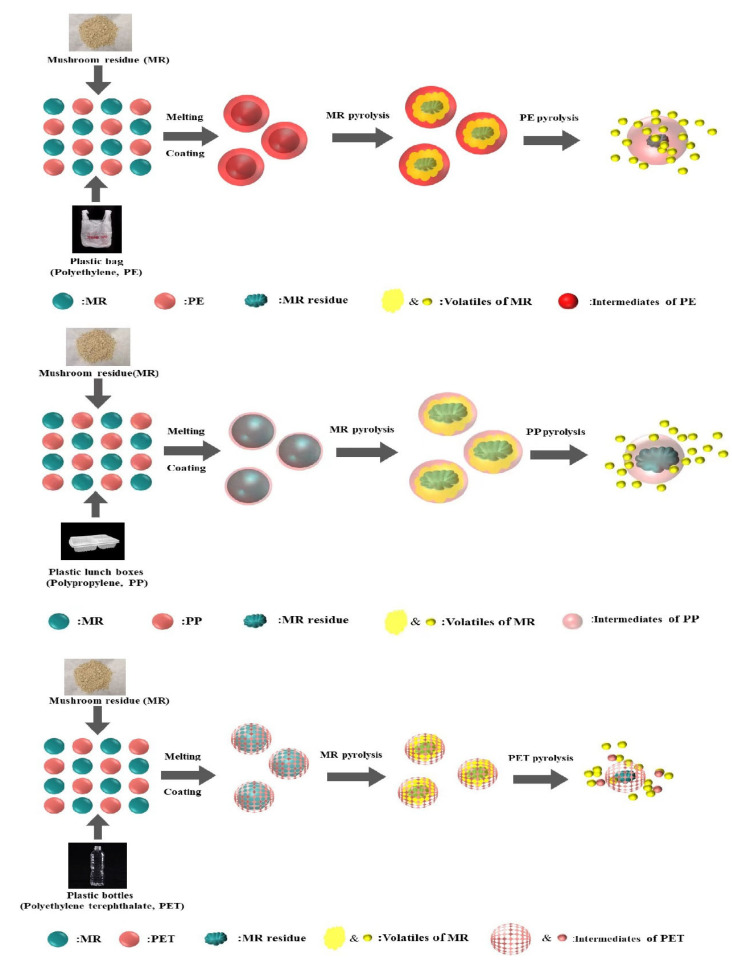
Interactions of MR and plastics according to ΔW variation.

**Figure 7 polymers-15-03824-f007:**
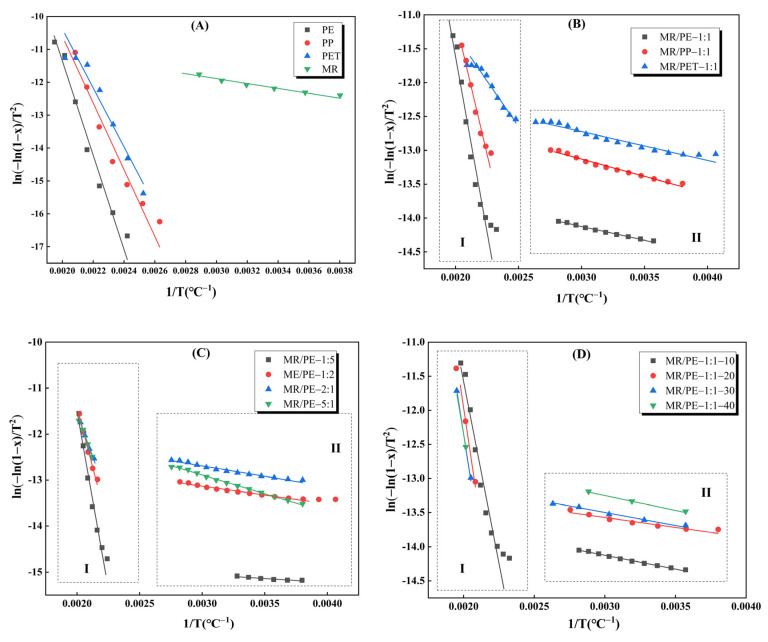
Plots of $\ln(-\ln\left(1-x\right)/T^{2})$ versus 1⁄T of samples, (**A**) MR, PE, PP and PET; (**B**) MR with the addition of PE, PP or PET; (**C**) MR/PE mixture in different MR to PE ratio; (**D**) MR/PE mixture under different heating rate. The I zone and the II zone in the subfigure represent the first main pyrolysis stage and the second main pyrolysis stage, respectively.

**Table 1 polymers-15-03824-t001:** Pyrolysis parameters for MR, PE, PP, PET, and their mixtures.

	Pyrolysis Range (°C)				Maximum Weight Loss Rate (°C)/Min		Peak Temperature (°C)	
	T_i1_ ^a^	T_f1_ ^b^	T_i2_ ^c^	T_f2_ ^d^	(dm1/dt)_max_ ^e^	(dm2/dt)_max_ ^f^	T_p1_ ^g^	T_p2_ ^h^
Single components								
MR-10 °C/min	201	497	-	-	-0.50	-	342	-
PE-10 °C/min	- ^i^	-	416	503	-	−3.39	-	480
PP-10 °C/min	-	-	385	492	-	−2.75	-	463
PET-10 °C/min	-	-	390	494	-	−2.11	-	446
Effect of plastic types								
MR/PE-1:1-10 °C/min	203	391	391	505	−0.08	−2.37	339	481
MR/PP-1:1-10 °C/min	228	377	377	496	−0.22	−1.59	341	467
MR/PET-1:1-10 °C/min	219	374	374	615	−0.26	−0.71	341	433
Effect of MR/PE ratio								
MR/PE-5:1-10 °C/min	205	410	380	500	−0.28	−1.16	340	478
MR/PE-2:1-10 °C/min	203	417	385	501	−0.27	−1.13	340	480
MR/PE-1:1-10 °C/min	203	391	391	505	−0.08	−2.37	339	481
MR/PE-1:2-10 °C/min	209	409	409	503	−0.20	−1.75	339	482
MR/PE-1:5-10 °C/min	266	399	399	508	−0.03	−2.89	340	485
Effect of heating rate								
MR/PE-1:1-10 °C/min	203	391	391	505	−0.08	−2.37	339	481
MR/PE-1:1-20 °C/min	212	401	401	515	−0.13	−1.63	350	491
MR/PE-1:1-30 °C/min	232	412	412	522	−0.14	−1.21	348	496
MR/PE-1:1-40 °C/min	236	419	419	526	−0.16	−0.86	362	498

^a^ Initial decomposition temperature of the first main pyrolysis stage; ^b^ final decomposition temperature of the first main pyrolysis stage; ^c^ initial decomposition temperature of the second main pyrolysis stage; ^d^ final decomposition temperature of the second main pyrolysis stage; ^e^ maximum weight loss rate of the first main pyrolysis stage; ^f^ maximum weight loss rate of the second main pyrolysis stage; ^g^ temperature of maximum weight loss rate of the first main pyrolysis stage; ^h^ temperature of maximum weight loss rate of the second main pyrolysis stage; ^i^ the item is unapplicable.

**Table 2 polymers-15-03824-t002:** Kinetic analysis parameters of samples.

	Heating Rate (°C/min)	Temperature Range (°C)	α (%)	E (kJ/mol)	A (min^−1^)	R^2^
Single components						
MR	10	263~360	24.84–67.17	6.18	$4{.74\;\times \;10}^{-1}$	0.9669
PE		413~513	9.75–99.50	119.05	$4{.59\;\times \;10}^{12}$	0.9742
PP		380~496	1.29–99.73	84.30	$1{.61\;\times \;10}^{9}$	0.9628
PET		396~496	3.23–95.72	74.38	$1{.61\;\times \;10}^{8}$	0.9529
Effect of plastic types						
MR/PE-1:1	10	280~363	4.55–9.91	3.33	$9{.76\;\times \;10}^{-3}$	0.9935
		430~513	12.19–95.86	79.19	$1{.53\;\times \;10}^{8}$	0.9448
MR/PP-1:1		263~363	9.15–25.86	4.25	$4{.70\;\times \;10}^{-2}$	0.9822
		430~496	31.88–93.34	55.84	$6{.09\;\times \;10}^{5}$	0.9551
MR/PET-1:1		246~380	12.16–39.07	3.54	$4{.57\;\times \;10}^{-2}$	0.9543
		396~480	41.99–84.00	19.96	$3{.29\;\times \;10}^{1}$	0.9578
Effect of MR/PE ratio						
MR/PE-5:1	10	263~363	8.90–32.79	6.79	$2{.39\;\times \;10}^{-1}$	0.9974
		463~496	48.60–86.90	61.91	$2{.08\;\times \;10}^{6}$	0.9931
MR/PE-2:1		263~363	14.49–36.83	3.78	$5{.47\;\times \;10}^{-2}$	0.9723
		460~496	48.76–87.38	56.06	$4{.44\;\times \;10}^{5}$	0.9820
MR/PE-1:1		280~363	4.55–9.91	3.33	$9{.76\;\times \;10}^{-3}$	0.9935
		430~513	12.19–95.86	79.19	$1{.53\;\times \;10}^{8}$	0.9448
MR/PE-1:2		246~363	8.60–24.80	2.86	$1{.90\;\times \;10}^{-2}$	0.9545
		460~496	37.07–90.80	83.51	$5{.75\;\times \;10}^{8}$	0.9834
MR/PE-1:5		263~313	1.75–2.16	1.64	$1{.05\;\times \;10}^{-3}$	0.9438
		446~500	7.87–95.54	123.87	$1{.26\;\times \;10}^{13}$	0.9714
Effect of heating rate						
MR/PE-1:1	10	280~363	4.55–9.91	3.33	$9{.76\;\times \;10}^{-3}$	0.9935
		430~513	12.19–95.86	79.19	$1{.53\;\times \;10}^{8}$	0.9448
MR/PE-1:1	20	263~363	7.13–17.19	2.42	$1{.78\;\times \;10}^{-2}$	0.9416
		463~513	19.66–95.02	93.55	$8{.31\;\times \;10}^{9}$	0.9742
MR/PE-1:1	30	280~380	8.52–20.18	3.17	$4{.93\;\times \;10}^{-2}$	0.9843
		463~513	25.56–88.81	75.70	$9{.58\;\times \;10}^{7}$	0.9410
MR/PE-1:1	40	280~363	10.35–23.72	3.92	$1{.38\;\times \;10}^{-1}$	0.9925
		480~517	39.26–91.75	86.72	$2{.16\;\times \;10}^{9}$	0.9882

## Data Availability

Data is contained within the article or Appendix A.

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
