# Peer review of "Thermal Behavior and Pyrolysis Kinetics of Mushroom Residue with the Introduction of Waste Plastics"

_polymers, 2023, doi:10.3390/polym15183824_

Round 1

Reviewer 1 Report

In my opinion, the authors present an adequate study on the degradation of mushroom residues mixed with different types of conventional plastics.

Some suggestions:

- It is not very clear what applications this type of mixtures or methodology could have. It is suggested to go deeper into this aspect.
- The sample preparation method ensures an intimate mixture between the components?. Has this been verified by e.g., FTIR?
- Correct the labeling on the X axis of Figure 5

It is acceptable

Author Response

In my opinion, the authors present an adequate study on the degradation of mushroom residues mixed with different types of conventional plastics.

Many thanks for your comments on our manuscript. Based on your suggestions, we have carefully revised the manuscript as possible as we can. We hope the revised version could satisfy the requirement of Polymers. Please check the yellow parts in the revised manuscript.

Q1: It is not very clear what applications this type of mixtures or methodology could have. It is suggested to go deeper into this aspect.

Answer: Indeed, further work is still needed to achieve the synergistic treatment of various solid wastes through co-pyrolysis. However, there have been numerous related research reports that have successfully obtained high-value chemicals, carbon materials, and fuels through co-pyrolysis [1-2]. We will continue to devote ourselves to this work, and once there are new discoveries, we will report them promptly.

References:

[1] Rajia Sultana, Uchhwas Banik, Pranab Kumar Nandy, Muhammad Nurul Huda, Mohammad Ismail. (2023). Bio-oil production from rubber seed cake via pyrolysis: Process parameter optimization and physicochemical characterization. Energy Conversion and Management: X, 20, 100429.

[2] Ramesh Potnuri, Chinta Sankar Rao, Dadi Venkata Surya, Abhishankar Kumar, Tanmay Basak. (2023). Utilizing support vector regression modeling to predict pyro product yields from microwave-assisted catalytic co-pyrolysis of biomass and waste plastics. Energy Conversion and Management, 292, 117387.

Q2: The sample preparation method ensures an intimate mixture between the components?. Has this been verified by e.g., FTIR?

Answer: Indeed, as you said, an intimate mixture is crucial for the co-pyrolysis. Based on our previous investigations including TG, FT-IR, proximate analysis and ultimate analysis (data not shown in the manuscript), we found that a grinding time of 10 minutes can ensure sufficient blending of the samples.

Q3: Correct the labeling on the X axis of Figure 5.

Answer: The labeling on the X axis of Fig. 5 has been revised. Please check the revised manuscript.

Reviewer 2 Report

Authors - Jiale Li, Tao Pu, Zhanghong Wang and Taoze Liu

Article – “Thermal Behavior and Pyrolysis Kinetics of Mushroom Residue with the Introduction of Waste Plastics”

This work is an important study that allows us to develop useful treatment and use of industrial and agricultural production residues. However, there are some small comments and questions.

1.    It is necessary to rewrite the Abstract. Based on the first sentence, it follows that agricultural and forestry waste will be subjected to co-pyrolysis. After that, it is unclear where the waste of polymer production comes from. In addition, the remains of edible mushrooms are a by-product of agriculture (see the first sentence).

2.    “Currently, the main methods to treat edible fungi residues mainly include incineration, composting, making animal feed, making flower substrate, and reusing to cultivate edible fungi.” References are required to this statement.

3.    Previously, the samples were dried at 60 C. Was it carried out under vacuum? At the same time, the first stage of pyrolysis begins with 30 C . Does this mean that the samples have not been dried? Why is the initial pyrolysis temperature of 30 C chosen?

4.    In the “Materials and methods” indicate the particle sizes of polymer powder, but do not indicate the particle sizes of mushroom residue powder. It is required to specify. How homogeneous is the powder mixture?

Author Response

This work is an important study that allows us to develop useful treatment and use of industrial and agricultural production residues. However, there are some small comments and questions.

Many thanks for your comments on our manuscript. Based on your suggestions, we have carefully revised the manuscript as possible as we can. We hope the revised version could satisfy the requirement of Polymers. Please check the yellow parts in the revised manuscript.

Q1: It is necessary to rewrite the Abstract. Based on the first sentence, it follows that agricultural and forestry waste will be subjected to co-pyrolysis. After that, it is unclear where the waste of polymer production comes from. In addition, the remains of edible mushrooms are a by-product of agriculture (see the first sentence).

Answer: According to your suggestions, we have made careful revisions to the description of the research background in the abstract. Please check the Abstract section in the revised manuscript.

Co-pyrolysis is considered a very promising technology for the treatment of solid wastes as it can rapidly realize the volume reduction of raw materials and obtain high value-added products. To realize the resource utilization of newly emerging solid wastes regarding to edible fungi residue and waste plastics, mushroom residue (MR), a representative of edible fungi residue, was co-pyrolyzed with waste plastic bags (PE), waste plastic lunch boxes (PP) and waste plastic bottles (PET).

Q2: “Currently, the main methods to treat edible fungi residues mainly include incineration, composting, making animal feed, making flower substrate, and reusing to cultivate edible fungi.” References are required to this statement.

Answer: The related references have been added. Please check the Introduction section in the revised manuscript.

Q3: Previously, the samples were dried at 60 C. Was it carried out under vacuum? At the same time, the first stage of pyrolysis begins with 30 C. Does this mean that the samples have not been dried? Why is the initial pyrolysis temperature of 30 C chosen?

Answer: The samples was dried in an oven under an air atmosphere for 12h. Then the obtained samples were directly used for further studies. Please check the Materials and methods section in the revised manuscript.

The initial temperature for thermogravimetric analysis is room temperature, not 30℃, and it may vary slightly with the ambient temperature. Our previous description was incorrect, and it has now been corrected. Please check the Materials and methods section in the revised manuscript.

Q4: In the “Materials and methods” indicate the particle sizes of polymer powder, but do not indicate the particle sizes of mushroom residue powder. It is required to specify. How homogeneous is the powder mixture?

Answer: The particle sizes of mushroom residue powder has added. Please check the Materials and methods section in the revised manuscript.

Indeed, a homogeneous and intimate mixture is crucial for the co-pyrolysis. Accordingly, a grinding time of 10 min was carried out, which has been proved enough to get a homogeneous and intimate mixture.